# Efficacy of superimposing neuromuscular electrical stimulation onto core stability exercise in patients with nonspecific low back pain: A study protocol for a randomized controlled trial

Yongzhong Li[1*¤a], Qian Fang[2¤b☉], Zhe Meng[1¤a‡], Xuan Li[1‡], Haixin Song[1☉], Jianhua Li[1☉]

**1** Department of Rehabilitation medicine, Sir Run Run Shaw Hospital, Zhejiang University School of Medicine, Hangzhou, Zhejiang, P.R. China, **2** Department of Rehabilitation medicine, Zhejiang Province Youth Hospital, Hangzhou, Zhejiang, P.R. China

☉ These authors contributed equally to this work.
‡ ZM and XL also contributed equally to this work.
¤aCurrent address: Department of Rehabilitation medicine, Sir Run Run Shaw Hospital, Zhejiang University School of Medicine, Hangzhou, 310016, P.R. China
¤bCurrent address: Department of Rehabilitation medicine, Zhejiang Province Youth Hospital, Hangzhou, 310016, P.R. China
* 162183@zju.edu.cn

## Abstract

### Introduction

Non-specific low back pain (NSLBP) is a prevalent condition affecting individuals worldwide, leading to significant disability and healthcare costs. Traditional treatment methods have shown limited efficacy, prompting the exploration of innovative approaches. Core stability exercise (CSE) has emerged as a promising rehabilitation strategy, yet optimal activation of local muscle systems remains to be fully understood. This trial aims to assess the efficacy of superimposing neuromuscular electrical stimulation (NMES) onto CSE for improving muscle activation, function, and proprioception in NSLBP patients.

### Methods and analysis

A total of 52 participants aged 18–60 years with NSLBP will be randomly allocated into two groups: (1) experimental group receiving NMES superimposed on CSE and (2) control group undergoing the same CSE with sham NMES. Interventions will occur three times per week for six weeks. The primary outcome measures will encompass surface electromyography (sEMG) to assess muscle activity and muscle activation timing. Secondary outcomes will include the evaluation of pain intensity using the Visual Analog Scale (VAS) and disability measured by the Oswestry Disability Index (ODI), as well as proprioception assessed through joint repositioning error (JRE) and muscle thickness evaluated via real-time ultrasound image (RUSI).

**Data availability statement:** No datasets were generated or analysed during the current study. All relevant data from this study will be made available upon study completion.

**Funding:** The author(s) received no specific funding for this work.

**Competing interests:** The authors have declared that no competing interests exist.

Data will be collected at baseline, after six weeks, and at a six-month follow-up. A mixed ANOVA will be employed to compare differences among groups and to analyze trends over time as well as interaction effects between treatment and time.

## Trial registration

**Trial registration number:** Chinese Clinical Trial Registry (ChiCTR2400092409)

## Introduction

Low back pain (LBP) is one of the most common global health issues, contributing to significant disability, societal burden, and healthcare costs. It is estimated that 75%–85% of individuals will experience at least one episode of LBP during their lifetime, a trend that has steadily increased over the past two decades [1]. As of 2017, the global prevalence of LBP was approximately 7.5%, equating to approximately 577 million individuals worldwide [2]. After excluding specific pathologies affecting the lumbar spine, about 90% of LBP cases cannot be attributed to a specific diagnosis and are thus classified as non-specific low back pain (NSLBP) [3]. Given the high prevalence and impact of NSLBP, it is crucial to develop and refine treatment strategies that can effectively improve patient outcomes.

One such approach is core stability exercise (CSE), which has emerged as a common intervention for LBP. CSE focuses on strengthening the muscles of the trunk and pelvis, enhancing movement patterns in the transverse and frontal planes. This approach not only prevents and reduces pain and disability but also improves spinal stability and flexibility, ultimately promoting the restoration of normal motor patterns [4]. However, despite its advantages, the long-term effects of CSE on pain relief and functional improvements remain limited. Issues such as poor patient adherence, time-consuming training sessions, and insufficient long-term effects persist [5].

A key challenge in treating NSLBP lies in addressing abnormal trunk muscle activation patterns. Patients with NSLBP exhibit abnormal activation of the trunk muscles, including insufficient and delayed activation, especially in the deep trunk muscles such as the transversus abdominis(TrA) and lumbar multifidus(LM) [6]. This dysfunction in deep muscle activation contributes to impaired spinal stability and movement patterns, which can exacerbate pain and disability. Conventional CSE, particularly in unloaded forms, may not adequately address these issues due to difficulties in effectively engaging these deep muscles through voluntary effort alone [7].

Neuromuscular electrical stimulation (NMES) offers a potential solution by directly stimulating the paraspinal muscles to enhance muscle contraction and motor unit recruitment. NMES, when combined with voluntary movements in exercise training, can further activate spinal reflex pathways and enhance central descending drive, improving muscle function and proprioception. This dual approach increases synaptic efficiency of somatosensory pathways and alters cortical activity in both primary and secondary somatosensory cortices, contributing to more effective motor control

and muscle recruitment [8,9]. Studies have shown that NMES superimposed on exercise training demonstrates significant positive effects on improving exercise performance and reducing time expenditure compared to using NMES alone or engaging in voluntary exercise training alone, while also providing additional neurological and physiological benefits [10,11].

Despite these promising findings, there is limited research on the effects of NMES superimposed on CSE in patients with NSLBP, and high-quality clinical randomized controlled trials are lacking. Furthermore, existing studies often do not address the complex interplay of motor control, proprioception, and cortical changes induced by this combined intervention. Therefore, the primary aim of this study is to investigate the efficacy of NMES superimposed on CSE in improving core muscle activation, proprioception, and spinal stability in NSLBP patients. We hypothesize that, compared to a standard CSE alone, the NMES-combined intervention group will induce greater core muscle activation, more effectively enhance trunk stability, reduce pain, and improve function.

## Methods

### Study design

This study will be a single-center, prospective, randomized, sham-controlled trial conducted at the outpatient rehabilitation center of the Sir Run Run Shaw Hospital, affiliated with Zhejiang University School of Medicine. A total of 52 participants will be recruited and randomly assigned in a 1:1 ratio to one of the following two groups: (a) experimental group, receiving NMES superimposed on CSE to enhance activity levels; (b) control group, receiving the same CSE with sham NMES. All participants in both groups will undergo 20-minute sessions, three times per week, for a 6-week intervention period. The study outcomes will be evaluated before (t0), six weeks after initiating interventions(t1), and six months (t2) after starting the protocol. The study protocol follows the standards described in the "Standard Protocol Items: Recommendations for Interventional Trials" guideline (S1 File) [12].

### Approval and registration of the study

The study design and procedures have been approved by the Institutional Review Ethics Committee of Sir Run Run Shaw Hospital, affiliated with Zhejiang University School of Medicine (Approval No. 2024–0285 and Date of review: April 25, 2024). The trial has also been prospectively registered with the Chinese Clinical Trial Registry (Registration No ChiCTR2400092409.) (S2 and S3 Files). We confirm that all methods will be conducted in accordance with the ethical standards of the Declaration of Helsinki. Participants will be informed about the study's purpose and procedures, and those meeting the inclusion criteria will be asked to sign a detailed consent form (S4 File).

### Participant involvement

None of the participants will be involved in the design, implementation, or dissemination plans of this study. Patients who meet the inclusion criteria will be recruited for the randomized controlled trial (RCT) through our outpatient department, posters, and social media platforms (from January 1st, 2025 to December 31th, 2026). Participants' personal data, such as names, ages, and weights, will be stored in a coded digital format within a secure database, accessible only to the research personnel responsible for randomization and blinding. Upon completion of the trial, participants will have access to their individual study results.

### Eligibility criteria

Inclusion Criteria: (a)Participants aged between 18 and 60 years [13]. (b)History of LBP for at least 3 months, without radiculopathy, specific spinal disorders, or radicular pain [14]. (c)Visual Analog Scale (VAS) score≤6. (d)No received CSE

or NMSE therapy within the past month. (e)Infrequent use of analgesics (less than 4 days per week). (f)No medication or medical conditions affecting muscle metabolism (e.g., corticosteroids). (g)Normal hearing and psychological status, with high adherence to participation. (h)All participants must sign the informed consent form. Exclusion Criteria are Pregnant or breastfeeding women [15]; Presence of severe underlying diseases such as cardiovascular, liver, or renal disorders; Participants with comorbid conditions like osteoarthritis, gout, tumors, acute trauma, or fractures that affect daily living capabilities [16]; History of spinal surgery; Contraindications for NMES (e.g., pacemakers, edema, sensory abnormalities, thromboembolism); Participants currently enrolled in other clinical trials for interventions targeting NSLBP.

## Sample size calculation

The sample size for this study is determined based on previous findings reported by Songjaroen et al [17]. The test statistic used to calculate the sample size was based on a mixed ANOVA. The calculation is based on the anticipated effect size for LM activation improvement, with a large effect size (Cohen's dz = 0.7), a statistical power of 80%, a significance level of 5%, and a standard deviation of 0.05 points. Using G*Power 3.1 software for the analysis, it is calculated that at least 43 participants are required. Considering an anticipated dropout rate of 20%, a total of 52 participants will be recruited for the study (26 in the experimental group and 26 in the control group).

## Randomization and blinding

After participants are enrolled in the study and have completed baseline assessments, an independent technician will generate a randomization sequence. The random sequence will be created using a random number generator (IBM SPSS Statistics V.25.0 software; IBM). The random allocation sequence will be sent to the therapists in sequentially numbered opaque envelopes [18]. The therapists will open the envelopes in sequence and assign participants, who meet the inclusion criteria, to the corresponding group. While the study assessors will be blinded to the group allocation of the participants, neither the therapists nor the participants can be blinded due to the nature of the interventions. The continued use of sham NMES for participants can play a key role in minimizing performance bias. Sham NMES provides a control condition that mimics the setup and experience of the intervention without delivering therapeutic electrical stimulation. This approach enables us to isolate the effects of active NMES on clinical outcomes (such as pain, proprioception, and muscle activation) by ensuring that any observed improvements are not simply due to placebo effects, psychological factors, or participants' awareness of receiving NMES treatment. To minimize bias, assessors, data managers, and analysts will remain blinded to group assignments during outcome assessments and data analysis. The study design and trial flowchart are illustrated in Figs 1 and 2.

## Strategies to increase treatment adherence

Assessors will maintain regular communication with participants throughout the study to monitor their progress and address any discomfort or challenges. Participants will be encouraged to promptly report any increase in pain, discomfort, or difficulty in performing exercises or tolerating NMES. If any issues arise, the research team will assess the individual's condition and make necessary adjustments to the treatment protocol. These adjustments may include modifying the intensity or duration of NMES, altering the difficulty level of exercises, or providing alternative movements to ensure proper execution while minimizing pain. If NMES settings prove intolerable, parameters such as frequency, intensity, or pulse duration may be adjusted to improve comfort without compromising effectiveness. Additionally, the progression of CSE will be personalized, allowing flexibility in exercise advancement based on each participant's physical capability. To enhance adherence, assessors will offer continuous guidance, encouragement, and regular check-ins, reinforcing motivation and addressing concerns in real time. By implementing these strategies, the study aims to optimize participant engagement, ensure a tolerable and effective intervention, and maximize the benefits of the treatment protocol.

| | STUDY PERIOD | | | |
|---|---|---|---|---|
| | Screening | Baseline | Six weeks | Follow up assessment |
| TIME POINT | -T1 | T0 | T1 | T2 |
| ENRONMENT: | | | | |
| Eligibility screen | X | | | |
| Informed consent | X | | | |
| Allocation | | X | | |
| INTERVENTION: | | | | |
| NMES+CSE | | | | |
| NMES sham+CSE | | | | |
| ASSESSMENTS: | | | | |
| Pain sensation | | X | X | X |
| Disability | | X | X | X |
| Proprioception | | X | X | X |
| Amplitude and onset timing (TrA,EO,ES,LM) | | X | X | X |
| Muscle activities (TrA,EO,ES,LM) | | X | X | X |
| Muscle thickness (TrA,LM) | | X | X | X |
| Use of medication | | X | X | X |

**Fig 1. Study design schedule.** CSE, core stability exercise; NMES, Neuromuscular electrical stimulation; TrA, transverse abdominis; EO, external oblique; IO, internal oblique; LM, lumbar multifidus; ES, erector spinae.

## Intervention measures

All participants will be randomly assigned to one of two groups, receiving treatment three times per week for a duration of 6 weeks. All treatments will be administered by therapists with over 10 years of clinical experience. The therapists must have undergone specialized training in CSE and passed clinical assessments to ensure consistency in the treatment provided to all patients. Building on previous studies, the therapists will demonstrate and explain a modified CSE training program to the participants in both groups (Table 1) [19]. The progression of CSE detailed in Table 1 represents a general framework for advancing exercise difficulty over the 6-week intervention period. The duration and structure of the CSE program were designed based on evidence from previous studies indicating the efficacy of similar protocols in patients with NSLBP [5,11,20]. However, we acknowledge that not all patients may progress at the same rate due to variability in pain levels, functional capacity, or motor control impairments. To accommodate individual differences, the exercise progression will be tailored to each patient's abilities and clinical presentation. The supervising therapist will evaluate the patient's pain, functional status, and motor performance at each session. Progression to subsequent levels will only occur when the patient demonstrates: Adequate pain tolerance during the current exercise level (i.e., no significant exacerbation of symptoms). Sufficient motor control and stability to safely perform the next level of exercises without compensatory movement patterns. During the treatment and observation periods of the study, if a participant's pain symptoms worsen, they will be instructed to take NSAIDs, specifically celecoxib (Pfizer Pharmaceuticals Ltd.), at a dose of 200 mg twice daily for two consecutive days. If the pain does not subside, the participant will be withdrawn from the study.

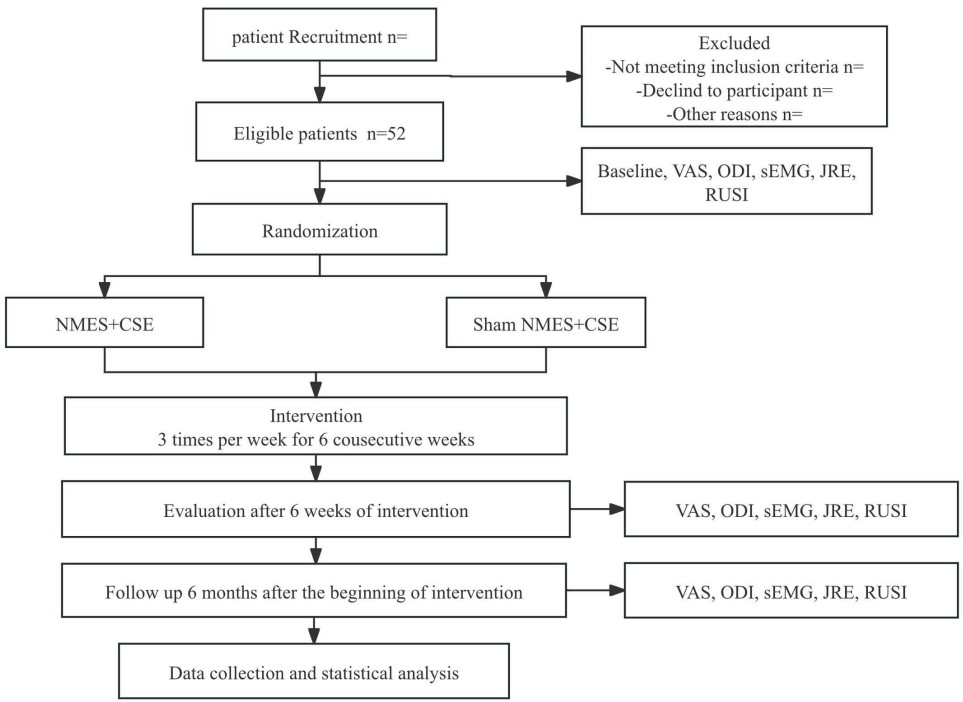

**Fig 2. Flow diagram of the study.** CSE, core stability exercise; NMES, Neuromuscular electrical stimulation; VAS, Visual Analog Scale; ODI, Oswestry Disability Index; JRE, joint repositioning error; RUSI, real-time ultrasound image.

**Table 1. Core stability exercise.**

| Week 1–2 | Week 3–4 | Week 5–6 |
|---|---|---|
| Supine Transverse Abdominis Contraction Training: Move the belly button towards the spine. | Supine Transverse Abdominis Contraction with Opposite Arm and Leg Lift (Not Touching the Ground). | Supine Transverse Abdominis Contraction with Double Leg Bridge. |
| Supine Transverse Abdominis Contraction with Heel Slide. | Supine Transverse Abdominis Contraction with Double Leg Bridge. | Supine Transverse Abdominis Contraction with Air Cycling Movement. |
| Supine Transverse Abdominis Contraction with Lower Limb Lift. | Supine Transverse Abdominis Contraction with Single Leg Bridge (Left Leg). | Supine Transverse Abdominis Contraction with Crunch (Hands Touching Knees). |
| Supine Transverse Abdominis Contraction with Contralateral Limb Elevation. | Supine Transverse Abdominis Contraction with Single Bridge (Right Leg). | Kneeling Hand-Knee Position with Transverse Abdominis Contraction While Raising the Opposite Upper and Lower Limbs. |
| Supine Transverse Abdominis Contraction with Double Leg Bridge. | Kneeling side bridge (left side). | Kneeling side bridge with knee extended (left side). |
| Kneeling Hand-Knee Position with Transverse Abdominis Contraction While Raising the Upper Limb. | Kneeling side bridge (right side). | Kneeling side bridge with knee extended (right side). |
| Kneeling Hand-Knee Position with Transverse Abdominis Contraction While Raising the Lower Limb. | Prone position transverse abdominal contraction with opposite limb raise. | Prone position transverse abdominal contraction with opposite limb raise. |
| Kneeling Hand-Knee Position with Transverse Abdominis Contraction While Simultaneously Raising the Contralateral Upper and Lower Limbs. | Kneeling Hand-Knee Position with Transverse Abdominis Contraction While Simultaneously Raising the Contralateral Upper and Lower Limbs. | Plank Exercise |

Participants in the experimental group will receive NMES using the EN-Stim4 device (ENRAF-NONIUS B.V., Bunnik, Netherlands) applied to the bilateral LM and TrA/internal oblique muscles(IO) (Fig 3). For the NMES configuration, the abdominal electrodes will be positioned as follows: the cathode (negative electrode) will be placed 1 cm above the iliac crest along the mid-axillary line, while the anode (positive electrode) will be located 2 cm medial to the anterior superior iliac spine. For the lumbar electrodes, the cathode (negative electrode) will be positioned approximately 2 cm lateral to the spinous processes of L4 and L5, with the anode (positive electrode) placed at a distance from the cathode to ensure optimal current flow through the lumbar region [21]. The electrodes used will be 5 cm × 5 cm surface hydrogel electrodes. The NMES used in this study will be applied with the following parameters: a frequency of 50 Hz, a 200 µs pulse width, and a biphasic symmetric waveform, which is effective for recruiting both superficial and deep muscle fibers [22]. The stimulation will follow a contraction-relaxation cycle comprising an ascent phase of 1 second, a contraction phase of 4 seconds, a descent phase of 1 second, and a resting phase of 6 seconds, lasting a total of 20 minutes. The current intensity will be adjusted to achieve the maximum muscle contraction without causing discomfort, such as burning sensations or severe cramping pain. NMES intensity is adjusted weekly based on participant feedback and tolerability, as well as to match improvements in muscle strength and activation capacity. Intensity increases are guided by the participant's perception of a "strong but comfortable" stimulation level and monitored through visible muscle contractions. CSE will be performed during the contraction phase of the electrical stimulation and will cease during the resting phase. These exercises will include abdominal activation training in supine, prone, and quadruped positions, as well as bridge exercises and planks (Table 1). Each session will consist of 8 different exercises, performing 1 set of 10 repetitions per exercise, lasting 20 minutes per session, once per day, 3 days per week [23]. Participants in the control group will undergo the same CSE and NMES treatment, but the NMES current intensity will be set to a level that only elicits minimal sensory perception in the participants (generally below 5 mA). This ensures that while the participants experience the same treatment protocol, they do not receive the therapeutic benefits of NMES, allowing for an effective comparison between the experimental and control groups.

## Primary outcome measures

Surface Electromyography (sEMG) Muscle Activity: Muscle activity will be recorded using the MegaWin ME6000-T8 surface electromyography system (Finland). The system will be set to manual recording mode, utilizing low-pass bipolar

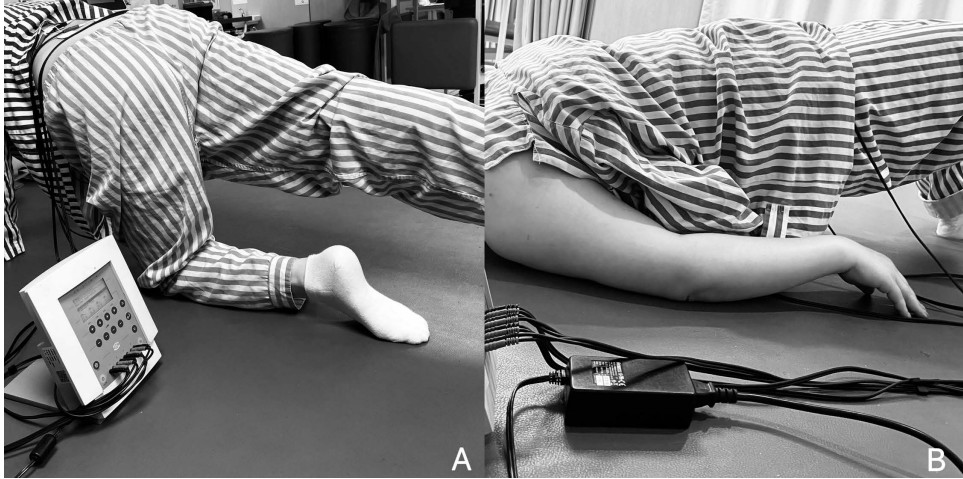

**Fig 3. Examples of NMES superimposed on CSE:** A) Quadruped position with transverse abdominal contraction while simultaneously raising the contralateral upper and lower limbs; B) Supine position with transverse abdominal contraction accompanied by a double bridge.

analog signals with a 32-bit microcomputer. The instrument sensitivity will be set to 1 μV, with a filtering range of 8–500 Hz and a sampling frequency of 1000 Hz. The measurements will be conducted using a differential preamplifier and a free raw data measurement method. Standardized disposable electrodes will be used to eliminate the possibility of reuse. The connection of the electrodes to the electromyography device will be adjusted to ensure that it does not impede the participants' movements and to prevent the disconnection of the surface electrodes or data line connectors due to tension. The electrode spacing will be set to 2.5 cm and positioned parallel to the muscle fibers of the following muscles: external oblique(EO) (approximately 15 cm lateral to the umbilicus), TrA/IO (2 cm superior and medial to the anterior superior iliac spine), erector spinae(ES) (2 cm lateral to the spinous processes of L1), LM (2 cm lateral to the spinous processes of L5), and anterior deltoid (two fingerbreadths from the anterior aspect of the acromion) [24]. All electrodes will be placed on the more severely affected side of NSLBP patients. The reference electrode will be connected to the iliac crest. The EMG data will be measured for 5 seconds. After excluding the first and last second, the root mean square (RMS) values of the 3 seconds of sEMG signal will be calculated [25].

Measurement of Muscle Activation Timing: To assess the latency of trunk muscle activation, participants will perform a lifting task while standing quietly, which introduces an inherent perturbation to the trunk and requires anticipatory postural adjustments (APA) [24]. The electrode placements will be the same as those used for muscle activity monitoring. Participants will stand with their elbows fully extended at 0°, shoulders flexed at 40°, and grasp the handles of a 5 kg box placed on a table in front of them. They will be instructed to lift the box to a shoulder flexion angle of 90° as quickly as possible and maintain this position for 3 seconds. After familiarizing themselves with the movement, participants will complete three repetitions of the lifting task, with a 1-minute interval between each trial. If any trunk or pelvic motion is observed during the trial, that attempt will be considered a failure. For sEMG processing, the procedures for manually selecting activation timings are as follows: Raw sEMG signals will first be filtered using a band-pass filter (20–450 Hz) to remove noise and artifacts. The baseline sEMG levels will be defined by calculating the RMS value of a 500 ms resting period before the onset of muscle activation. Muscle activation will be identified when the sEMG signal exceeds 3 standard deviations (SD) above the baseline RMS for a minimum duration of 50 ms. If necessary, trained assessors will visually inspect and manually adjust the activation timing to account for signal variations and artifacts, ensuring consistency across trials. The relative difference in onset activation times between each trunk muscle and the anterior deltoid will be calculated using the following formula: Relative Onset Activation Time Difference = Onset Activation Time of Trunk Muscle - Onset Activation Time of Anterior Deltoid (ms) [26]. A positive value indicates that the target trunk muscle is activated after the anterior deltoid. The average onset time from the three repetitions for each trunk muscle will be computed and used as the basis for data analysis.

## Secondary outcome measures

Pain: Self-reported pain will be assessed using the VAS [27]. The VAS is a tool used to quantify the level of pain experienced by participants during the study period, with scores ranging from 0 to 10, where 0 indicates "no pain" and 10 indicates "unbearable pain."

Disability: The Oswestry Disability Index (ODI) will be used to measure functional disability. This scale consists of 10 sections, each scored from 0 to 5 points, with higher scores indicating greater disability. The relative value will be expressed as the total score divided by the total possible score, multiplied by 100%. Participants will select the option that best describes their level of functional disability on the day of assessment.

Real-time ultrasound image (RUSI) Assessment of Muscle Thickness: The evaluation of muscle thickness will be conducted using the SONIMAGE HS1 (KONICA MINOLTA, Shanghai, China) equipped with a 65 mm convex array transducer (C5-2 type, frequency 5 MHz). For the abdominal muscles, participants will be positioned supine with their knees flexed at 90° and hips flexed at approximately 45°. The transducer will be placed gently at the intersection of a horizontal line passing through the umbilicus and a vertical line passing through the anterior superior iliac spine to visualize the TrA/IO.

Measurements will be taken at the end of expiration. For the LM, participants will be positioned prone with a pillow placed under the abdomen to minimize lumbar lordosis. Using the sacrum as a reference, the probe will be positioned longitudinally over the spinous processes of L3, L4, and L5 and moved laterally to observe the LM at these levels to measure muscle thickness [28]. The muscle activation ratio will be calculated as follows:

$$Muscle\ Activation\ Ratio = \frac{Contracted\ Thickness}{Relaxed\ Thickness}$$

For the TrA, the preferred activation ratio will be calculated using the formula:

$$Preferred\ Activation\ Ratio = \frac{TrA\ Contracted\ Thickness}{(TrA + IO + EO)\ Contracted\ Thickness} - \frac{TrA\ Relaxed\ Thickness}{(TrA + IO + EO)\ Relaxed\ Thickness}$$

A higher value of the preferred activation ratio indicates a relatively greater change in the contracted thickness of the TrA, while a lower value suggests a relatively greater change in the contracted thickness of the EO and IO muscles [19].

Proprioception Testing: Proprioception will be assessed using joint repositioning error(JRE) as the primary outcome measure. Participants will be seated with their feet flat on the ground and hands resting on their thighs. The examiner will guide the participant into the neutral position of the lumbar spine. The measurement will start at the sacral level 1 (S1), with the center of a 10 cm measuring tape positioned at the measurement starting point. A laser pointer will be placed 50 cm behind the participant, aimed at the center of the 10 cm measuring tape located at S1. The examiner will instruct the participant to remember the target position, then perform two maximum anterior pelvic tilts and posterior pelvic tilts, holding each position for 5 seconds before returning to the neutral target position. The deviation from the starting point will be measured in centimeters as the JRE [29]. Prior to the assessment, participants will undergo two practice trials to familiarize themselves with the testing procedure. A total of three tests will be conducted, with a 1-minute rest between each test. The average of the absolute errors from the three measurements will be calculated to represent the magnitude of the error.

## Data collection and management

Data will be collected at baseline, at week 6, and during the 6-month follow-up. Throughout the training period, all data will be recorded and stored in case report forms (CRFs). Researchers will also contact participants via phone to ensure they complete follow-up assessments. If a participant fails to complete the assessment within two days of the scheduled date, they will receive a reminder phone call. At the conclusion of the trial, all research data will be extracted from the CRFs into Excel spreadsheets by two data managers who are blind to the group allocation. These data managers will have completed rigorous data monitoring training and will input the real-time data into the Chinese Clinical Trial Registry. After the data have been double-checked for accuracy, statisticians will conduct statistical analyses on the data. Since this study does not involve a drug trial, and the sponsors or funders will not have access to the raw data, there will be no establishment of a data monitoring committee, nor are there plans for trial audits. Additionally, due to the very low risk of adverse events and other unexpected impacts, this study will not conduct interim analyses, and no stopping guidelines have been formulated.

## Statistical analysis

Statistical analyses will be performed using SPSS Version 22.0 software (IBM). Continuous variables that conform to a normal distribution will be expressed as means ± standard deviations, while non-normally distributed continuous variables will be presented as medians (interquartile range). Categorical variables will be reported as proportions and rates. Baseline comparisons for continuous variables will be conducted using t-tests or non-parametric tests as appropriate, while baseline comparisons for categorical variables will utilize chi-square tests. For measurements taken at multiple time points, a mixed

ANOVA will be employed to compare differences among groups and to analyze trends over time as well as interaction effects between treatment and time. In the presence of significant interaction effects between time and group, post hoc tests will be conducted with Bonferroni adjustments for multiple comparisons. The significance level will be set at 0.05 (two-tailed).

## Discussion

The treatment of NSLBP poses significant challenges, with a substantial proportion of patients continuing to experience symptoms despite receiving treatment. This ongoing issue may stem from a limited understanding of the mechanisms that initiate and sustain pain. The unique aspect of this study is the simultaneous use of NMES with active CSE, providing a quantitative analysis of the synergistic effects on the improvement of muscle structure, function, and proprioception in NSLBP. Objective assessment methods will comprehensively evaluate the activation patterns, thickness changes, and motor control ability of the deep lumbar muscles. Data will be collected at baseline, at the end of the 6-week intervention, and at a 6-month follow-up to reflect both short-term and mid-term effects, addressing the gap in long-term effect assessment in existing research. Literature indicates that 6 weeks of CSE can significantly improve pain levels and motor function in NSLBP patients, with notable improvements in core muscle thickness and activation capacity [5,6,13]. The muscle remodeling effects of NMES typically manifest within 4–6 weeks [30], and extending the intervention period further may increase patient dropout rates, reducing the feasibility of the study. The selected pulse width of 200 μs for NMES is consistent with previous research on optimizing parameters for deep lumbar muscle NMES, ensuring effective muscle recruitment while maintaining patient comfort [22,23]. This parameter minimizes the risk of activating superficial muscles, which may contribute little to core stability in patients with NSLBP. Additionally, the individualized progression of exercises ensures that each patient reaches optimal engagement in the treatment, while minimizing the risk of symptom exacerbation or maladaptive movement patterns.

This study employs sEMG technology to assess the activation patterns of deep core muscles such as the LM and TrA. Its high sampling rate and sensitivity help capture dynamic muscle changes, particularly in quantifying muscle activation timing differences, which further reveal motor control impairments in patients [31,32]. The anticipated improvements in sEMG outcomes include increased muscle activation, with NMES enhancing motor unit recruitment, leading to more consistent activation of the TrA and LM during movement. This is particularly important for individuals with NSLBP, as these muscles often fail to activate effectively during functional tasks. The protocol also aims to improve motor control through neuromuscular system retraining, enhancing proprioceptive feedback and muscle coordination. This will result in better lumbar stability during movement, reducing the risk of injury. Over time, improvements in these sEMG outcomes are expected to correlate with functional benefits, such as reduced pain, increased stability, and improved movement patterns, allowing individuals to perform daily activities with less discomfort and greater ease, potentially reducing pain recurrence.

Furthermore, RUSI will be used to precisely measure the thickness and dynamic changes of core muscles, with quantification of the thickness changes in the TrA, IO,EO and LM effectively reflecting muscle remodeling and the structural effects of the NMES+CSE intervention. JRE will be used to quantify proprioceptive function, serving as a key tool to assess sensory input dysfunction in NSLBP patients. Studies have shown that improvements in proprioception are significantly associated with pain relief and enhanced motor function [29]. This study innovatively combines NMES superimposed on CSE interventions with JRE assessments to quantify lumbar proprioception recovery and further explore the relationship between sensory input and motor output. Our goal is to provide robust evidence to inform the design of a more efficient and practical treatment for NSLBP patients.

### Dissemination

The findings of this study will be disseminated through publication in peer-reviewed international journals and presentations at national and international conferences. Participants involved in the study will also receive the research results via telephone or email.

## Supporting information

**S1 File. SPIRIT—checklist.**
(DOCX)

**S2 File. Study protocol approved Chinese.**
(DOCX)

**S3 File. Study protocol approved English.**
(DOCX)

**S4 File. Informed consent form.**
(DOCX)

## Acknowledgments

We would like to thank the patients and staff at the Rehabilitation Medicine Department of Sir Run Run Shaw Hospital, Zhejiang University School of Medicine, for their contributions to piloting the intervention.

## Author contributions

**Conceptualization:** Yongzhong Li, Haixin Song.

**Data curation:** Qian Fang, Zhe Meng, Xuan Li.

**Formal analysis:** Yongzhong Li.

**Investigation:** Yongzhong Li, Xuan Li.

**Methodology:** Yongzhong Li, Qian Fang, Zhe Meng, Xuan Li, Haixin Song.

**Project administration:** Qian Fang, Jianhua Li.

**Software:** Qian Fang, Zhe Meng.

**Supervision:** Yongzhong Li, Haixin Song, Jianhua Li.

**Validation:** Haixin Song, Jianhua Li.

**Writing – original draft:** Yongzhong Li.

**Writing – review & editing:** Yongzhong Li.

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
