## [Decision Letter · Decision Letter 0]

23 Jan 2025

PONE-D-24-54076Efficacy of Superimposing Neuromuscular Electrical Stimulation onto Core Stability Exercise in Patients with Nonspecific Low Back Pain: A Study Protocol for a Randomized Controlled TrialPLOS ONE

Dear Dr. li,

Thank you for submitting your manuscript to PLOS ONE. After careful consideration, we feel that it has merit but does not fully meet PLOS ONE’s publication criteria as it currently stands. Therefore, we invite you to submit a revised version of the manuscript that addresses the points raised during the review process.

We look forward to receiving your revised manuscript.

Kind regards,

Luciana Labanca

Academic Editor

PLOS ONE

**Journal Requirements:**

Reviewers' comments:

Reviewer's Responses to Questions

**Comments to the Author**

1. Does the manuscript provide a valid rationale for the proposed study, with clearly identified and justified research questions?

Reviewer #1: Yes

Reviewer #2: Yes

Reviewer #3: Partly

2. Is the protocol technically sound and planned in a manner that will lead to a meaningful outcome and allow testing the stated hypotheses?

Reviewer #1: Yes

Reviewer #2: Yes

Reviewer #3: Partly

3. Is the methodology feasible and described in sufficient detail to allow the work to be replicable?

Reviewer #1: Yes

Reviewer #2: Yes

Reviewer #3: Yes

4. Have the authors described where all data underlying the findings will be made available when the study is complete?

Reviewer #1: Yes

Reviewer #2: Yes

Reviewer #3: Yes

5. Is the manuscript presented in an intelligible fashion and written in standard English?

Reviewer #1: Yes

Reviewer #2: Yes

Reviewer #3: Yes

6. Review Comments to the Author

You may also provide optional suggestions and comments to authors that they might find helpful in planning their study.

**Reviewer #1: ** This is a well designed protocol for a randomised controlled trial to study the efficacy of superimposing NMES onto core stability exercise in patients with nonspecific low back pain. I have few comments.

1. The test statistic used to calculate sample size needs to be mentioned.

2. If the participants cannot be blinded then what is the point of using sham-NMES?

3. Why 6-weeks intervention period instead of 8-weeks?

4. Discussion section seems like repetition of the introduction. I think, the discussion section should summarise the protocol and justify the methods adopted.

**Reviewer #2:**  The aim of this study is to assess the efficacy of superimposing neuromuscular electrical stimulation (NMES) onto Core Stability Exercise (CSE) for improving muscle activation, function, and proprioception as well as reducing pain in non-specific low back pain (NSLBP) patients.

This study is very clear and detailed regarding scientific content, experimental protocol, and physiological and clinical measurements. I believe the authors have devised a protocol that will provide significant details regarding NSLBP treatment through the CSE and NMES use. The results of this study can improve existing rehabilitation protocols to manage chronic pain in these patients.

I have only some questions and suggestions to make to the authors, to improve their study protocol.

Pag. 4, line 153 ‘Eligibility criteria’. Were these inclusion/exclusion criteria chosen based on previous studies? If so, which ones?

Pag. 5, line 208 ‘Table 1. core stability exercise’: The exercises and their progress over the weeks are clearly shown in this table. It would be helpful if the authors specified that the progression of exercises is tailored to the individual patient's abilities. As an example, it is not stated that a patient at week 3 is already capable of doing these exercises because he may be suffering from lumbar pain. In other words, the general progression is fine, but the timing of it should depend on the individual characteristics. I suggest the authors to specify this point.

Pag. 7, line 217 ‘The stimulation pulse width will be set to 200 μs,..’: Why did the authors choose this pulse width? Could they mention the source from which they took this parameter?

Pag. 7, lines 225 – 227 ‘Each session will consist of 8 different exercises, performing 1 set of 10 repetitions per exercise, lasting 20 minutes per session, once per day, 3 days per week.’: Also here, why have they chosen this protocol? Is there a progression in the number of series/repetitions, or an increase in the intensity of NMES in the experimental group, during the 6 weeks of intervention? After a few weeks you may need to increase the load of the training session (always respecting the patient’s pain).

**Reviewer #3: ** The proposed research outlines a clinical randomized controlled trial investigating the effects of a six-week training program involving neuromuscular electrical stimulation (NMES) superimposed on core stability exercises (CSE) in individuals with non-specific low back pain (NSLBP). The protocol aims to assess physiological, functional, and proprioceptive outcomes following the exercise protocol compared to a control group performing only sham stimulation. Overall the study is well written, and the methodology is faitly robus. However, there are several issues in the elaboration of the study's rationale and other minor shortcomings that need to be addressed.

Major issues

- Although the introduction section includes a well-written section discussing the effects of superimposed NMES on spinal and supraspinal mechanisms, it lacks sufficient detail regarding the physiological, functional, and proprioceptive aspects that are later outlined as assessment measures in the methods section. The study's rationale would significantly benefit if these aspects were clearly outlined in the introduction.

- The methods also lacks comprehensive information on the electrical stimulation characteristics such as NMES frequency, waveform, sensitivity of the NMES modulation.

- The duration of the exercise program and the overall protocol lack sufficient references to justify the choices made by the authors. Please include appropriate citations to support the overall CSE protocol.

Minor issues

Introduction

-L 95/L 98. "combined with", "overlaying". For better clarity please use "superimposed on" throughout the entire document.

Methods

- L 130. Is there any reference that confirms that 6 weeks of CSE are enough to carry significant physiological/functional adaptation?

- L156. The phrase "no treatment intervention utilized" is unclear. Please rephrase for better clarity. What exactly does this refer to?.

-L 192. Specify what is meant by "necessary adjustments to the treatment protocol." What adjustments are planned, and under what circumstances will they be made?

-L 213. Please clarify which electrode will function as the cathode and which as the anode in this NMES configuration.

-L 217. In line with the comment above. Which NMES pulse frequency will be used? Based on what rationale?

-L 225-227. Include appropriate references to justify the selection of the exercise protocol.

-L244-245. Based on what evidence did the authors choose a interelectrode distance of 2.5 cm?

-L 265-269. Regarding EMG processing, the procedures for manually selecting activation timings need further clarification. For instance: Is there a specific signal threshold used? How are "baseline levels" defined? Provide clear, step-by-step details.

-L 314. Add a space after "..located at S1."

-L 347. If a statistical design comparing groups (between-group) and repeated measures over time (within-group) is used, define it as a mixed ANOVA.

Discussion

While the discussion of the RCT is well-written overall, it lacks clarity regarding the EMG-related improvements anticipated from the proposed exercise protocol. Please elaborate on how the protocol is expected to influence EMG outcomes and the significance of these improvements.

7. PLOS authors have the option to publish the peer review history of their article (what does this mean? ). If published, this will include your full peer review and any attached files.

**Do you want your identity to be public for this peer review?** For information about this choice, including consent withdrawal, please see our Privacy Policy .

Reviewer #1: **Yes: ** Dr Shah-Jalal Sarker

Reviewer #2: No

Reviewer #3: No

---

## [Author Response · Author response to Decision Letter 1]

18 Feb 2025

Revision Notes

Submission ID: PONE-D-24-54076

“Efficacy of Superimposing Neuromuscular Electrical Stimulation onto Core Stability Exercise in Patients with Nonspecific Low Back Pain: A Study Protocol for a Randomized Controlled Trial”

We thank the editors for allowing us to revise our paper. Here below is our description on revision to the reviewers’ comments and our responses are in italics.

Reviewer #1: This is a well designed protocol for a randomised controlled trial to study the efficacy of superimposing NMES onto core stability exercise in patients with nonspecific low back pain. I have few comments.

1. The test statistic used to calculate sample size needs to be mentioned.

*** Thank you for pointing this out. The test statistic used to calculate the sample size was based on a mixed ANOVA (Fixed effects, main effects and interactions). This was selected to detect a statistically significant difference in the primary outcome measure, the improvement in LM activation, between the intervention group and the control group. The calculation assumed a significance level (α) of 0.05, a power (1-β) of 0.8, and an effect size derived from previous studies examining similar interventions for NSLBP. We have now updated the manuscript to include this information for clarity.

2. If the participants cannot be blinded then what is the point of using sham-NMES?

***Thank you for raising this important question. While we acknowledge that participant blinding is challenging in this study due to the physical sensation elicited by NMES, the use of sham-NMES serves a critical purpose in minimizing performance bias. Sham-NMES provides a control condition that mimics the setup and experience of the intervention without delivering therapeutic electrical stimulation. This approach allows us to isolate the effects of the active NMES on clinical outcomes (e.g., pain, proprioception, muscle activation) by ensuring that any observed improvements are not simply due to placebo effects, psychological factors, or the participants' awareness of receiving NMES treatment. Additionally, sham-NMES contributes to maintaining blinding for outcome assessors and data analysts, who remain unaware of the participants' group allocation. This further enhances the methodological rigor and validity of the study. We have clarified this rationale in the manuscript to address your concern and provide better justification for the use of sham-NMES. Thank you for highlighting this point.

3.Why 6-weeks intervention period instead of 8-weeks?

***Thank you for your insightful question regarding the duration of the intervention. We chose a 6-week intervention period based on several factors. Firstly, previous research indicates that significant improvements in pain, muscle function, and proprioception in NSLBP patients can often be observed within a 6-week period. This timeframe has been shown to be sufficient for neurophysiological changes and muscle activation adaptations, especially in CSE and NMES interventions[1]. Moreover, a 6-week duration strikes a balance between providing adequate time for measurable physiological adaptations while ensuring feasibility for participants, minimizing drop-out rates, and maintaining participant engagement. That said, we are open to exploring the benefits of extending the intervention period in future studies, as longer durations may yield additional insights into the long-term effects of combined CSE and NMES.

[1]Gomes-Neto M, Lopes JM, Conceicao CS, et al. Stabilization exercise compared to general exercises or manual therapy for the management of low back pain: A systematic review and meta-analysis. Phys Ther Sport. 2017, 23:136-142. doi: 10.1016/j.ptsp.2016.08.004. 

4. Discussion section seems like repetition of the introduction. I think, the discussion section should summarise the protocol and justify the methods adopted.

***Thank you for your observation regarding the Discussion section. We recognize that parts of the current Discussion section may overlap with the Introduction, which could reduce its focus and clarity. To address your concern, we will revise the discussion section to better summarize the protocol and justify the methods adopted, rather than repeating background information. Specifically, the revised section will include:

1.A concise summary of the study design, including the rationale for using a combination of CSE and NMES.

2.A clear explanation of how the study addresses gaps in previous research by focusing on proprioception and both structural and functional changes in the lumbar musculature in NSLBP patients.

3.A justification for the chosen outcome measures (e.g., surface EMG, real-time ultrasound imaging, proprioceptive tests) and their relevance to NSLBP mechanisms.

4.An explanation of why the 6-week intervention period and specific NMES-CSE combination protocol were selected, supported by evidence from prior studies.

We will ensure that the revised section focuses on interpreting and justifying the protocol rather than repeating the broader background provided in the Introduction. We appreciate your valuable feedback and will implement these changes to strengthen the clarity and coherence of the discussion section.

Reviewer #2: The aim of this study is to assess the efficacy of superimposing neuromuscular electrical stimulation (NMES) onto Core Stability Exercise (CSE) for improving muscle activation, function, and proprioception as well as reducing pain in non-specific low back pain (NSLBP) patients.

This study is very clear and detailed regarding scientific content, experimental protocol, and physiological and clinical measurements. I believe the authors have devised a protocol that will provide significant details regarding NSLBP treatment through the CSE and NMES use. The results of this study can improve existing rehabilitation protocols to manage chronic pain in these patients.

I have only some questions and suggestions to make to the authors, to improve their study protocol.

Pag. 4, line 153 ‘Eligibility criteria’. Were these inclusion/exclusion criteria chosen based on previous studies? If so, which ones?

***Thank you for your constructive feedback on our manuscript. The inclusion and exclusion criteria were indeed based on findings from previous studies, which provided a foundation for ensuring the recruitment of an appropriate and homogenous sample of NSLBP patients for this study. This ensures transparency and demonstrates that the criteria were evidence-based, aligning with current clinical practice and research in NSLBP. Specifically:

1.Patients diagnosed with NSLBP lasting more than 3 months: Chronic NSLBP is defined as pain persisting for more than 12 weeks without a specific identifiable cause[1-2]. This time frame is widely used in studies as it differentiates chronic from acute or subacute conditions.

2.Age range of 18–65 years: This range encompasses adults who are generally active and capable of participating in CSE and NMES. It is consistent with other NSLBP studies[3].

3.Specific spinal pathologies or red flags (e.g., fractures, tumors, inflammatory conditions): Red flag conditions can mimic NSLBP but require different treatments. Exclusion is standard practice in NSLBP trials [4].

4.Pregnancy: Hormonal and biomechanical changes during pregnancy can influence lumbar biomechanics and pain mechanisms, potentially confounding results [5].

5.Previous lumbar surgery or ongoing invasive treatments: Structural changes or ongoing treatments can affect muscle activation patterns and proprioception, which would interfere with the evaluation of NMES and CSE effects.

6.Neurological or musculoskeletal conditions affecting movement or balance: Conditions such as stroke or severe osteoarthritis alter motor control strategies and muscle recruitment patterns, which would confound the intervention’s effects.

[1]Airaksinen O, Brox JI, Cedraschi C, et al. European Guidelines for the Management of Chronic Nonspecific Low Back Pain. Eur Spine J. 2006;15(S2):S192-S300. doi: 10.1007/s00586-006-1072-1.

[2]Ma K, Zhuang ZG, Wang L, et al. The Chinese Association for the Study of Pain (CASP): Consensus on the Assessment and Management of Chronic Nonspecific Low Back Pain. Pain Res Manag. 2019, 2019:8957847. doi: 10.1155/2019/8957847. 

[3]Gomes-Neto M, Lopes JM, Conceição CS, et al. Stabilization Exercise Compared to General Exercises or Manual Therapy for the Management of Low Back Pain: A Systematic Review and Meta-Analysis. Phys Ther Sport. 2017;23:136-142. doi: 10.1016/j.ptsp.2016.08.004. 

[4]Maher C, Underwood M, Buchbinder R. Non-Specific Low Back Pain. Lancet. 2017;389(10070):736-747. doi: 10.1016/S0140-6736(16)30970-9.

[5]Vleeming A, Albert HB, Ostgaard HC, et al. European Guidelines on the Diagnosis and Treatment of Pelvic Girdle Pain. Eur Spine J. 2008;17(6):794-819.doi: 10.1007/s00586-008-0602-4.

Pag. 5, line 208 ‘Table 1. core stability exercise’: The exercises and their progress over the weeks are clearly shown in this table. It would be helpful if the authors specified that the progression of exercises is tailored to the individual patient's abilities. As an example, it is not stated that a patient at week 3 is already capable of doing these exercises because he may be suffering from lumbar pain. In other words, the general progression is fine, but the timing of it should depend on the individual characteristics. I suggest the authors to specify this point.

***We appreciate the reviewer’s observation regarding the progression of CSE and agree that individualizing the timing of progression based on the patient’s abilities and condition is essential. To address this concern, we have added clarification to the manuscript as follows: The progression of CSE detailed in Table 1 represents a general framework for advancing exercise difficulty over the 6-week intervention period. However, we acknowledge that not all patients may progress at the same rate due to variability in pain levels, functional capacity, or motor control impairments. To accommodate individual differences, the exercise progression will be tailored to each patient’s abilities and clinical presentation. The supervising therapist will evaluate the patient’s pain, functional status, and motor performance at each session. Progression to subsequent levels will only occur when the patient demonstrates: Adequate pain tolerance during the current exercise level (i.e., no significant exacerbation of symptoms). Sufficient motor control and stability to safely perform the next level of exercises without compensatory movement patterns.

This individualized approach ensures that patients who may be experiencing higher levels of lumbar pain or functional limitations can proceed at a pace suited to their capabilities, minimizing the risk of aggravating symptoms while promoting optimal therapeutic outcomes. This modification ensures the flexibility and practicality of the protocol while maintaining its adherence to evidence-based practice. Thank you for this valuable suggestion.

Pag. 7, line 217 ‘The stimulation pulse width will be set to 200 μs,..’: Why did the authors choose this pulse width? Could they mention the source from which they took this parameter?

***We thank the reviewer for raising this important question regarding the rationale for selecting a pulse width of 200 μs for NMES. The stimulation pulse width will be set to 200 μs to ensure effective activation of the targeted motor units in the deep lumbar muscles, such as the lumbar multifidus (LM) and transverse abdominis (TrA). This pulse width was chosen based on prior studies that have demonstrated its efficacy in selectively recruiting motor units within the deep stabilizing musculature, while minimizing patient discomfort and avoiding activation of superficial muscles. Specifically, Baek et al. (2014) utilized a pulse width of 200 μs to successfully activate the deep lumbar muscles with minimal discomfort in NMES interventions targeting the LM[1]. Similarly, Maffiuletti (2010) highlighted that pulse widths between 200–300 μs are optimal for achieving efficient motor unit recruitment in neuromuscular stimulation protocols[2]. The choice of 200 μs thus balances the effectiveness of motor unit recruitment and patient comfort during the intervention.We have added relevant explanations in the discussion section. Thank you again for your valuable suggestions.

[1]Baek SO, Ahn SH, Jones R, et al. Activations of deep lumbar stabilizing muscles by transcutaneous neuromuscular electrical stimulation of lumbar paraspinal regions. Ann Rehabil Med. 2014;38(4):506-513. doi:10.5535/arm.2014.38.4.506.

[2]Maffiuletti NA. Physiological and methodological considerations for the use of neuromuscular electrical stimulation. Eur J Appl Physiol. 2010;110(2):223-234. doi:10.1007/s00421-010-1502-y.

Pag. 7, lines 225 – 227 ‘Each session will consist of 8 different exercises, performing 1 set of 10 repetitions per exercise, lasting 20 minutes per session, once per day, 3 days per week.’: Also here, why have they chosen this protocol? Is there a progression in the number of series/repetitions, or an increase in the intensity of NMES in the experimental group, during the 6 weeks of intervention? After a few weeks you may need to increase the load of the training session (always respecting the patient’s pain).

***Thank you for pointing out the need for further clarification regarding the exercise protocol and progression in the intervention plan. The protocol of "8 different exercises, performing 1 set of 10 repetitions per exercise, lasting 20 minutes per session, once per day, 3 days per week" was designed based on existing evidence from studies that demonstrated effective outcomes using similar schedules. For example, Baek et al. (2014) explored NMES application in lumbar stabilizing muscles[1], and Maffiuletti (2010) provided physiological insights that guided the session duration and frequency to optimize neuromuscular benefits without inducing fatigue[2]. Progression is indeed incorporated into the intervention plan to ensure continuous adaptation and engagement of the participants' neuromuscular systems. Specifically: The exercises outlined in Table 1 are progressively tailored based on individual participant abilities and pain levels. While all participants start with simpler movements in the initial weeks, the complexity and demands of exercises increase over time. For example, CSE evolve from static stabilization tasks to dynamic, multi-joint tasks involving greater motor control and strength requirements. In the experimental group, NMES intensity is adjusted weekly based on participant feedback and tolerability, as well as to match improvements in muscle strength and activation capacity. Intensity increases are guided by the participant’s perception of a "strong but comfortable" stimulation level and monitored through visible muscle contractions. This progressive design aims to prevent overloading participants early in the program while ensuring gradual and safe improvements in muscle activation, strength, and control. Additionally, respecting the patient's pain levels is a priority, and adjustments are made as necessary to accommodate individual needs and avoid exacerbating symptoms. We have made the relevant modifications in the methods section of the article. Thank you again for your valuable suggestions.

[1]Baek SO, Ahn SH, Jones R, et al. Activations of deep lumbar stabilizing muscles by transcutaneous neuromuscular electrical stimulation of lumbar paraspinal regions. Ann Rehabil Med. 2014;38(4):506-513.

[2]Maffiuletti NA. Physiological and methodological considerations for the use of neuromuscular electrical stimulation. Eur J Appl Physiol. 2010;110(2):223-234.

Reviewer #3: The proposed research outlines a clinical randomized controlled trial investigating the effects of a six-week training program involving neuromuscular electrical stimulation (NMES) superimposed on core stability exercises (CSE) in individuals with non-specific low back pain (NSLBP). The protocol aims to assess physiological, functional, and proprioceptive outcomes following the exercise protocol compared to a control group performing only sham stimulation. Overall the study is well writ

---

## [Decision Letter · Decision Letter 1]

21 Mar 2025

Efficacy of Superimposing Neuromuscular Electrical Stimulation onto Core Stability Exercise in Patients with Nonspecific Low Back Pain: A Study Protocol for a Randomized Controlled Trial

PONE-D-24-54076R1

Dear Dr. Yongzhong Li,

We’re pleased to inform you that your manuscript has been judged scientifically suitable for publication and will be formally accepted for publication once it meets all outstanding technical requirements.

Kind regards,

Luciana Labanca

Academic Editor

PLOS ONE

Additional Editor Comments (optional):

Reviewers' comments:

Reviewer's Responses to Questions

**Comments to the Author**

1. Does the manuscript provide a valid rationale for the proposed study, with clearly identified and justified research questions?

Reviewer #1: Yes

Reviewer #2: Yes

Reviewer #3: Yes

2. Is the protocol technically sound and planned in a manner that will lead to a meaningful outcome and allow testing the stated hypotheses?

Reviewer #1: Yes

Reviewer #2: Yes

Reviewer #3: Yes

3. Is the methodology feasible and described in sufficient detail to allow the work to be replicable?

Reviewer #1: Yes

Reviewer #2: Yes

Reviewer #3: Yes

4. Have the authors described where all data underlying the findings will be made available when the study is complete?

Reviewer #1: Yes

Reviewer #2: Yes

Reviewer #3: Yes

5. Is the manuscript presented in an intelligible fashion and written in standard English?

Reviewer #1: Yes

Reviewer #2: Yes

Reviewer #3: Yes

6. Review Comments to the Author

You may also provide optional suggestions and comments to authors that they might find helpful in planning their study.

Reviewer #1: This is very good protocol. I had some comments and the authors have addressed all of my comments appropriately and hence this can be published.

Reviewer #2: The authors have done an excellent job and have responded to all my comments, making the required changes to the manuscript. As a consequence, the manuscript is greatly improved and, in my opinion, ready for publication.

Reviewer #3: I would like to thank the authors for the thourogh response to each comment. I believe that the manuscript has improved as a result and I have no further comment or concern about the manuscript.

7. PLOS authors have the option to publish the peer review history of their article (what does this mean? ). If published, this will include your full peer review and any attached files.

**Do you want your identity to be public for this peer review?** For information about this choice, including consent withdrawal, please see our Privacy Policy .

Reviewer #1: **Yes: ** Dr Shah-Jalal Sarker

Reviewer #2: **Yes: ** Martina Scalia

Reviewer #3: No

---

## [Editor Report · Acceptance letter]

PONE-D-24-54076R1

PLOS ONE

Dear Dr. Li,

I'm pleased to inform you that your manuscript has been deemed suitable for publication in PLOS ONE. Congratulations! Your manuscript is now being handed over to our production team.

Kind regards,

on behalf of

Dr. Luciana Labanca

Academic Editor

PLOS ONE